# Development of Woolly Hair and Hairlessness in a CRISPR−Engineered Mutant Mouse Model with KRT71 Mutations

**DOI:** 10.3390/cells12131781

**Published:** 2023-07-05

**Authors:** Tao Zhang, Hongwu Yao, Hejun Wang, Tingting Sui

**Affiliations:** Key Laboratory of Zoonosis Research, Ministry of Education, Institute of Zoonosis, College of Veterinary Medicine, Jilin University, Changchun 130062, China; ztao19@mails.jlu.edu.cn (T.Z.); yaohw0515@163.com (H.Y.); wanghj9920@mails.jlu.edu.cn (H.W.)

**Keywords:** CRISPR/Cas9, KRT71, woolly hair, hairlessness, mouse model

## Abstract

Hypotrichosis simplex (HS) and woolly hair (WH) are rare and monogenic disorders of hair loss. HS, characterized by a diffuse loss of hair, usually begins in early childhood and progresses into adulthood. WH displays strong coiled hair involving a localized area of the scalp or covering the entire side. Mutations in the keratin K71(KRT71) gene have been reported to underlie HS and WH. Here, we report the generation of a mouse model of HS and WH by the co−injection of Cas9 mRNA and sgRNA, targeting exon6 into mouse zygotes. The Krt71−knockout (KO) mice displayed the typical phenotypes, including Krt71 protein expression deletion and curly hair in their full body. Moreover, we found that mice in 3–5 weeks showed a new phenomenon of the complete shedding of hair, which was similar to nude mice. However, we discovered that the mice exhibited no immune deficiency, which was a typical feature of nude mice. To our knowledge, this novel mouse model generated by the CRISPR/Cas9 system mimicked woolly hair and could be valuable for hair disorder studies.

## 1. Introduction

Hair disorders in children are associated with the expression of several genetic, cutaneous, and systemic disorders. The mammalian hair follicle (HF) has an important organic structure that produces hair with several distinct cell layers [1], including the inner root sheath (IRS), the companion layer, and the outer root sheath. The IRS is composed of three layers: the IRS cuticle, the Huxley layer, and the Henle layer. Hair anomalies may exhibit changes in color, density, length, and structure in the hair shaft [2].

Woolly hair (WH) and Hypotrichosis simplex (HS) comprise a group of rare, monogenic disorders regarding hair loss. WH is an autosomal−dominant WH (ADWH) or autosomal−recessive WH (ARWH) that is characterized by the presence of less than the normal amount of hair and abnormal hair follicles and shafts that are thin and atrophic, which is considered to be a hair growth deficiency [3]. Recent advances in molecular genetics have led to the identification of numerous genes that are expressed in the WH, including LPAR6 [4], LIPH [5], KRT71 [6], and KRT74 [7].

HS is inherited in an autosomal−dominant or recessive manner [8], which is characterized by diffuse hairless [9]. Mutations in five of these genes—CDSN (MIM 602593), APCDD1 (MIM 607479), SNRPE (MIM 128260), KRT71 (MIM 608245), and KRT74 (MIM 608248)—are associated with autosomal dominant forms. In addition, mutations in these genes have been identified as the pathogenic reason in less than 20 HR cases/families, thus accounting for only a small proportion of all HS cases.

Keratins play an important role in the structural component of the HF and form keratin intermediate filaments (KIFs) through heterodimerization between type I (acidic) and type II (basic to neutral) keratins [10,11]. The KRT71 gene has been mapped on the human chromosome 12q13, and the precise expression patterns of IRS have been characterized in detail [12].

At present, the curly coat type is a relatively common trait in animals, including mice [13,14], dogs [15], rats [16], and cats [17], and is linked to a wavy−coat phenotype. Krt71 mutant mice display a wavy pelage and curvy vibrissae, which are inherited in an autosomal dominant manner [13]. Portuguese water dogs, carrying a particular allele of Krt71, exhibit curly versus straight hair15. Rex rats also display curly hair with a Krt71 mutation [16]. Sphynx and Devon Rex cats carrying Krt71 mutations are responsible for the curly/wavy phenotypes [17]. However, the etiology of many WH and HS cases remains unexplained.

In this study, we established a novel Krt71 mouse model through the cytoplasm microinjection of Cas9 mRNA and a single guide RNA (sgRNA). These clustered regularly interspaced short palindromic repeats of (CRISPR)/Cas9 knockout (KO) mice showed curly hair and nudity in 3–5 weeks. This novel Krt71−KO mouse model could be a valuable resource for mechanism studies on the control of hair growth and differentiation.

## 2. Materials and Methods

### 2.1. Animals and Ethics Statement

The ICR mice used in this study were maintained at the Laboratory Animal Center of Jilin University. All experiments involving mice in this study were performed in accordance with the guide of the Animal Care and Use Committee of Jilin University.

### 2.2. CRISPR/Cas9 sgRNA Preparation, Embryo Microinjection and Embryo Transfer

The CRISPR/Cas9 sgRNA preparation, embryo microinjection, and embryo transfer were performed as previously described [18]. Briefly, mixtures of sgRNA (25 ng/μL) and Cas9 (100 ng/μL) were injected into embryos and were followed by their transfer into the oviduct of the recipient mother.

### 2.3. Mutation Detection in Mice by PCR and Sequencing

The genomic DNA from Krt71−KO and WT mice were extracted from a small piece of the toe by the TIANamp Genomic DNA Kit (TIANGEN, Beijing, China) according to the manufacturer’s instructions. The sgRNA target site was amplified by a PCR using the primer (Forward, 5′–GTATGAGGAGATTGCCCTGAAG–3′; Reverse, 5′–AGAGTGAGTAGAGAGGGAAGTG–3′). The PCR products were gel purified and cloned into the pGM−T vector (Tiangen, Beijing, China). The clones were sequenced and analyzed by Snapgene2.3.2 (Boston, MA, USA).

### 2.4. RNA Extraction and Quantitative RT–PCR

The total RNA was extracted from the skin of Krt71–KO and WT mice via the TRNzol–A+ reagent (Tiangen, Beijing, China), and the first–strand cDNA was synthesized using the cDNA first–strand synthesis kit (Tiangen, Beijing, China) according to the manufacturer’s instruction as previously described [19]. The cDNA was used to examine the expression of the Krt71 gene. The primers were used for RT–PCR in Appendix A. The qRT–PCR was performed by an SYBR Green I Real–Time PCR Kit (Tiangen, Beijing, China), and the 2^−ΔΔCT^ formula was used to determine and analyze gene expression while GAPDH was used for normalization. All experiments were repeated three times for each gene, and the data were expressed as the mean ± SEM. 

### 2.5. Body Weight and Survival Curve

The body weight and survival of age–matched Krt71–KO and WT mice were recorded for each week. All data were expressed as the mean ± c, and at least 6 individuals had used determinations in all experiments.

### 2.6. Flow Cytometry

Approximately 1 mL of peripheral blood was collected from WT and Krt71–KO mice into an EDTA–coated tube. The red blood cells were lysed using a Red Blood Cell Lysis Buffer (C3702, Beyotime, Shanghai, China) as instructed by the manufacturer, and antibody incubation was performed as described in previous work [20]. Briefly, single cells were stained with a premixed FITC Rat Anti–Mouse CD4 antibody (553046, BD Biosciences, Shanghai, China) and PE Rat Anti–Mouse CD8a antibody (553032, BD Biosciences), which used a Facs buffer for 30 min at 4 °C, avoiding direct light, and was then washed with 500 μL of a cold Facs buffer for 5 min. After being washed again, the red blood cells were resuspended in 200 μL of a Facs buffer. The suspended cells were analyzed by flow cytometry using a MoFlo Astrios cell sorter (BD Biosciences, Shanghai, China). The fluorescence–activated cell sorting data were analyzed using the FlowJo_V10_CL software.

### 2.7. Western Blotting

Western blot analysis was performed on protein lysates purified from mouse skin. Briefly, the skins were homogenized in a PBS buffer with the protease inhibitor cocktail. Protein concentrations were determined using a BCA protein quantification kit (Beyotime Biotechnology, Shanghai, China). In total, 30 μg of the total protein was suspended in the SDS sample buffer. The primary antibodies against Krt71 (1:1000, Genetex, Irvine, CA, USA) and GAPDH (1:2000, proteintech, Wuhan, China) were used. These were then incubated with the appropriate HRP–conjugated antibody (1:2000, Beyotime Biotechnology, Shanghai, China), which was used as an internal control. Protein bands were visualized by an ECL reagent (Meilunbio, Dalian, China).

### 2.8. Histology Analysis

Haematoxylin and eosin (HE) staining was performed as previously described [21]. The skin tissues were collected from Krt71–KO and WT mice (3 months of age). Then, these tissues were fixed in 4% paraformaldehyde, embedded in paraffin wax, sectioned, and mounted on slides. The 5-μm sections were cut for H&E and imaged with a Nikon TS100 microscope.

### 2.9. Scanning Electron Microscopy

Scanning electron microscopy was performed as previously described [22]. Briefly, hair from the beard of Krt71–KO and WT mice was attached to specimen stubs using stick conductive tabs, which were sputter–coated with gold and imaged by an S–3400N scanning electron microscope (Hitachi, Tokyo, Japan).

### 2.10. Statistical Analysis

The data are presented as the mean ± SEM. Data from multiple–groups were determined by unpaired Student’s *t*-test for two–group comparisons and one–way ANOVA with Bonferroni’s post–tests for multiple group comparisons. A *p*-value less than 0.05 was considered significant. Statistical analyses were performed using Prism 8 (GraphPad 8).

## 3. Results

### 3.1. Generation of Krt71 Mice the CRISPR/Cas9 System

To disrupt the function of Krt71, we designed a sgRNA targeting exon 6 for the mouse Krt71 gene (Figure 1A). The target site is shown in Figure 1A. Then, 126 injected zygotes were transferred into the oviducts of two surrogate mice. All surrogates were pregnant to term and gave birth to six live pups. The genomic DNA from each pup was extracted and tested for Krt71 mutations via a PCR, which was then analyzed by Sanger sequencing (Figure 1B). In total, five of the six (83.3%) newborn pups carried a Krt71 mutation (Figure 1C).To examine off–target effects in Krt71–KO mice, we predicted sequences similar to sgRNAs through an online website and selected the five most likely potential off–target sites (POTs) before these sites were detected by a PCR and Sanger sequencing. There were no off–target mutations at potential off–target sites (POTs) in Krt71 mutant mice (Appendix A). These results demonstrated that mutations in Krt71 could be achieved via the CRISPR/Cas9 system with high efficiency in mice.Additionally, no significant differences in body weight were found between newborn Krt71–KO mice and WT mice (Figure 1D). In addition, the survival rate of Krt71–KO mice was consistent with that of the WT mice (Figure 1E).

### 3.2. Phenotype Analysis of Krt71 Knockout Mice

Previous studies have reported that three affected members of a three–generation Japanese family segregated autosomal dominant woolly hair/hypotrichosis [6]. As expected, the 4–week–old Krt71–KO mice exhibited a visibly curly fur phenotype, including a beard, in comparison with age–matched WT mice (Figure 2A). Subsequently, we detected the mRNA expression level of Krt71 in knockout mice. The Krt71 mRNA expression level displayed a significant decrease in both the back and the abdomen of the Krt71–KO mice by qPCR (Figure 2C). In addition, we also further confirmed this by RT–PCR (Figure 2B). These results were also confirmed by Western blotting in both the back and the abdomen at the protein level (Figure 2D). In addition, the predicted 3D models showed a disrupted GADD45G protein structure (Figure 2E). These results suggested that the reduced expression mediated by NMD (nonsense–mediated mRNA decay, NMD) contributed to the generation of woolly hair/hypotrichosis in Krt71–KO mice. Therefore, we successfully constructed Krt71–KO mice models by CRISPR/Cas9.

Krt71 plays a central role in hair formation, which is an essential component of keratin intermediate filaments in the inner root sheath (IRS) of the hair follicle. In addition, some paralog genes Krt25, Krt27, Krt72, Krt75, and Krt85, which also participate in the formation of keratin intermediate filaments in the IRS, could be related to Woolly Hair, Autosomal Recessive 3, and Hypotrichosis 8, which were significantly decreased in Krt71–KO mice (Figure 3A–E). Krt25 and Krt27, located on the human chromosome 17q12 and encoding the members of the type I (acidic) keratin family, led to the formation of a keratin intermediate filament (KIF) network by forming a heterodimer with type II keratins. Thus, we guessed that the knockdown of the Krt71 expression influenced the expression of these corresponding molecular chaperones. For the same reason that Krt72, Krt75, and Krt85 were mapped on human chromosome 12q13 as Krt71, we considered that the destruction of the Krt71 gene loci in Krt71–KO mice impaired the expression of other genes since these two enhancers were found to exist in the KRT71 gene in human [23]. Previous studies have shown that MZF1, as a transcription factor, negatively regulates Krt71 gene expression by identifying the fragment of the Krt71 promoter [24]. We found that MZF1 expression increased by nearly −30 fold (Figure 3F). We speculated that the Krt71 mRNA fragment involved in NMD could further activate the expression of MZF1. More than that, LPAR6 is highly expressed in hair follicles, especially IRS, thus indicating the crucial role it plays in hair follicle development and hair growth and its association with Hypotrichosis 8 and Familial Woolly Hair Syndrome [25]. As shown in Figure 3G, the expression of LPAR6 mRNA was significantly improved in Krt71–KO mice. Furthermore, we found that TGF–α also increased dramatically to ~6-fold in Krt71–KO mice (Figure 3H). In terms of the above results, Krt71–KO mice not only reduced the expression of mRNA in keratin family genes but also increased the expression of other genes, which were all related to hair follicle development and influenced hair growth. We conjectured that an unknown compensation mechanism was inevitable in the process of hair development.

To further characterize the phenotype of Krt71–KO mice, we performed H&E staining of the back and abdominal skin sections in three different genotypes of KO mice. As shown in Figure 4A,B, H&E staining indicated that, compared with the WT mice, Krt71–KO mice exhibited hairy roots that were curved in all different genotypes of mice. These results indicate that Krt71–KO mice demonstrated significantly woolly hair.

### 3.3. Structural Changes of Beards in Krt71–KO Mice

Previous studies have demonstrated the occurrence of hair shafts that have been damaged in Krt71 mutations individuals. To determine the hair shafts of beards in Krt71–KO mice, we selected 6–week–old mice for scanning electron microscope analyses. As shown in Figure 5A, the Krt71–KO mice exhibited an apparent beard bend and were shorter than WT mice. The scanning electron microscope analyses showed that the beard hair shaft was altered, rough, and broken, and the texture change was unclear in Krt71–KO mice when compared with WT mice (Figure 5B).

In addition, to further determine the effect of Krt71 knockout on beard growth, we counted the beard length and amount. We found that the average whisker length in Krt71–KO mice was much shorter than that in WT mice (Figure 5C). We measured the whisker length on one side of the mice and found that approximately 86.7% of whiskers in Krt71–KO mice were <15 cm in length, while 60% in WT mice had whiskers >15 cm in length (Figure 5D).

### 3.4. Nude Appearance of Krt71–KO Mice

Regardless of the mutation type, Krt71–KO mice showed the nude phenotype at 3–4 weeks old (Appendix A), and the mice gradually regained their hair as they became older. In addition, we also found that Krt71–KO mice restored their normal hair density in development with age (Appendix A). The hair growth in Krt71–KO mice exhibited a cyclic pattern, which was similar to NuRabbits [26]. Therefore, to test whether Krt71–KO mice were similar to NuRabbits that were immunodeficient, we collected peripheral blood from the central ocular veins of mice who were 6 weeks old and analyzed these for populations of circulating lymphocytes. By flow cytometry, there was an apparent decrease in the percentage of T lymphocytes in Krt71–KO mice compared with WT mice (Appendix A and Figure 6A). With regards to the T cell subsets, populations of CD8–positive cells declined, whereas they had no significant differences, and CD4–positive T cells dramatically decreased from 20% to 13.9% (Figure 6A,B). However, on a postmortem examination, there were no marked reductions or absences in the thymus of Krt71 and WT mice (Appendix A). In Krt71–KO mice, body weights and thymus weights were both not significantly lower than that of WT mice (Figure 6D,E). We detected T cell subsets of the thymus and found that there were no differences between WT and Krt71–KO mice (Appendix A). Interestingly, previous studies have found that nude–mice exhibit the prominent up–regulation of keratin (Krt23, −73, −82, −16, −17) [27]. Furthermore, a significantly reduced Foxn1 mRNA expression was observed in the Krt71–KO mice compared with the WT mice (Figure 6F). Thus, our results indicate that Foxn1 may have a negative correlation with Krt71 and other keratins in mice.

## 4. Discussion

Keratin 71, encoded by the Krt71 gene, plays a central role in hair formation, which is an essential component of keratin intermediate filaments in the inner root sheath (IRS) of the hair follicle. In this study, we generated a novel mouse model for WH and HS via the zygote injection of Cas9 mRNA and a sgRNA targeting exon 6 of the Krt71 gene and demonstrated that Krt71–KO mice almost exhibited the woolly hair observed in human beings, including curly hair and a bent beard. To the best of our knowledge, this is the first report of a Krt71 gene mutant model that develops both woolly hair and hairless that also mimics the phenotype of hypotrichosis found in human ADWH [6].

Furthermore, our data showed that the significantly decreased expression of Krt71 caused the nude phenotype at 3–4 weeks old in Krt71–KO mice, which is a new characteristic in the models of mice. However, previous studies have found that Krt71 mutants only display wavy hair and curved whiskers, which are typical characters in whole knock–out models [13]. However, Krt71–KO mice still carried the more severe nude phenotype at 3–4 weeks old than other Krt71 mutants. We conjecture that the reason for this phenomenon was due to the different knockout results, which only deleted the asparagine codon at 139 or 140 in the Krt71Ca–9J mice and destructed the protein by the NMD mechanism in the –KO mice by CRISPR/Cas9. The hair phenotype was similar to Hypotrichosis 13 (HYPT13) with woolly hair in pedigree families [6]. Based on the report of one Japanese family, the symptoms improved with age, resulting in woolly hair that had an almost normal hair density, which was consistent with the Krt71–KO mice (Appendix A). Although the results showed that the expression of Foxn1 in the Krt71–KO mice was dramatically decreased compared to wild–type mice, we found that no apparent differences in thymus development and T cell subsets in the thymus were detected by flow cytometry.

In view of this, we inferred that the nude phenotype of Krt71–KO mice in the early stage was due to the knockout of the Krt71 gene, which resulted in generating a reduced density of hairs and the influence of the decreased expression of Foxn1. Thus, we constructed the Krt71–KO mice model using CRISPR/Cas9 technology which possessed a similar phenotype with HYPT13 and ADWH in humans. Therefore, the HYPT13 and ADWH models are valuable for studying hair development and disease treatment for elucidating the mechanisms underlying the normal process of hair development in humans.

Taken together, the Krt71–KO mouse model had a close resemblance to its human counterpart. This model could facilitate basic research to provide an ideal animal model and understand the mechanism of hair disorders.

## Figures and Tables

**Figure 1 cells-12-01781-f001:**
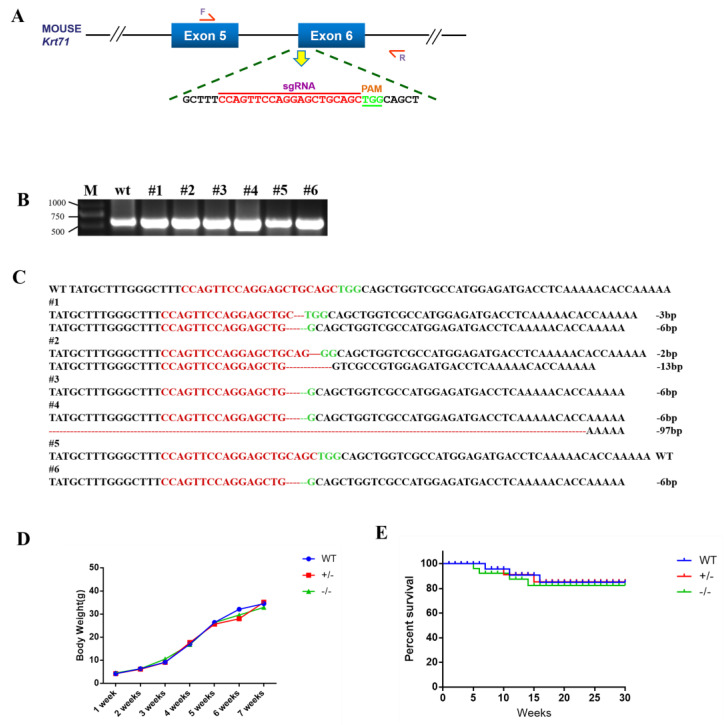
Generation of Krt71–KO mice by the CRISPR/Cas9 system. (**A**) Schematic diagram of sgRNA target site located in exon 6 of the mouse Krt71 locus. (**B**) Mutation detection by PCR in pubs 1–6; M, D2000. (**C**) Mutation detection by T–cloning and Sanger sequencing. PAM sites are highlighted in green; the target sequence is shown in red; deletions (−) of T–cloning sequence; WT, wild–type control. (**D**) Body–weight comparison of Krt71–KO and WT mice from newborn to 12 weeks. (**E**) Survival curves for male Krt71–KO mice.

**Figure 2 cells-12-01781-f002:**
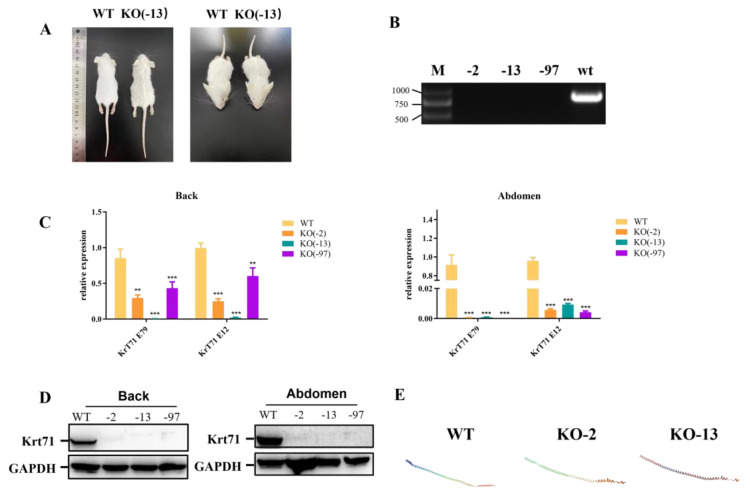
Phenotype characterization of Krt71–KO mice. (**A**) The gross performance of 12–week–old F1 Krt71–KO mice by photo. (**B**) The gene expression of Krt71 determined by RT–PCR. WT, WT control; KO, Krt71 gene knockout mice. (**C**) The gene expression of Krt71 determined by qRT–PCR in the back and abdomen. WT, WT control; KO, Krt71 gene knockout mice. (KO, *n* = 3; WT, *n* = 3). (**D**) Western blot analysis of Krt71 expression. (**E**) Computer modeling of the 3D structure of the Krt71 protein between WT and Krt71–KO (*n* = 3, ** *p* < 0.01, *** *p* < 0.001).

**Figure 3 cells-12-01781-f003:**
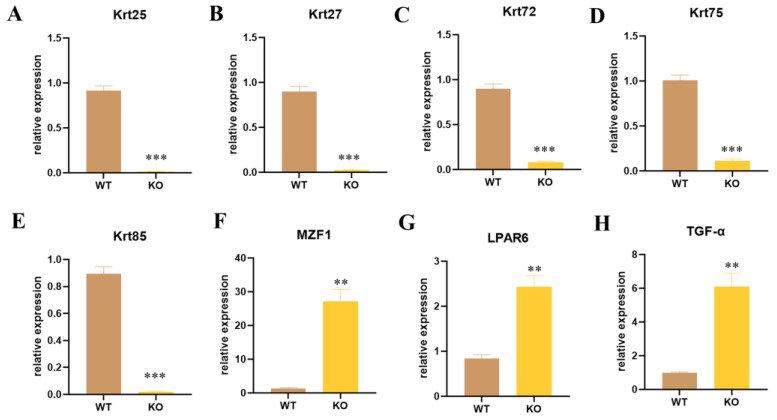
Expression level detection of different paralog genes and controlling genes in Krt71–KO mice. (**A**–**H**) The expressions of different genes were determined by qRT–PCR. WT, WT control; KO, Krt71 gene knockout mice (*n* = 3, ** *p* < 0.01, *** *p* < 0.001).

**Figure 4 cells-12-01781-f004:**
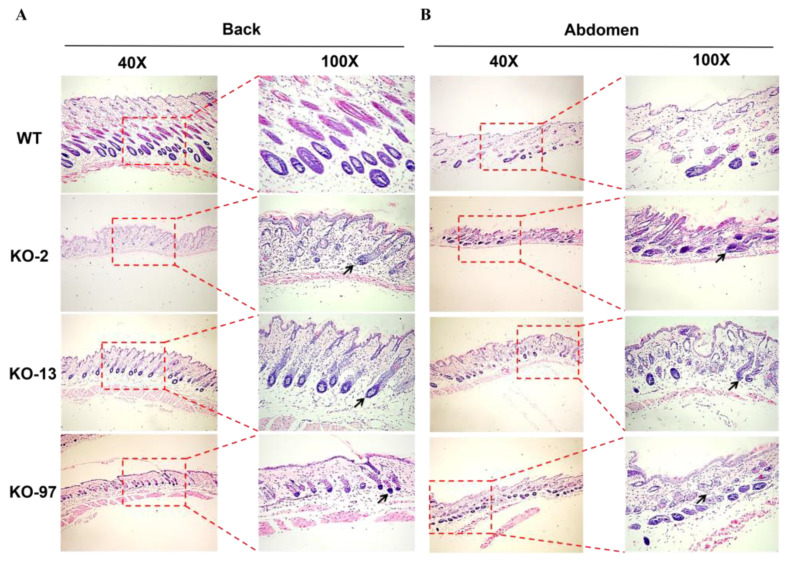
Skin and hair follicle changes in Krt71–KO mice. (**A**,**B**) H&E–staining of the skins from WT and Krt71–KO mice, showing hair bending (black arrows) in Krt71–KO mice.

**Figure 5 cells-12-01781-f005:**
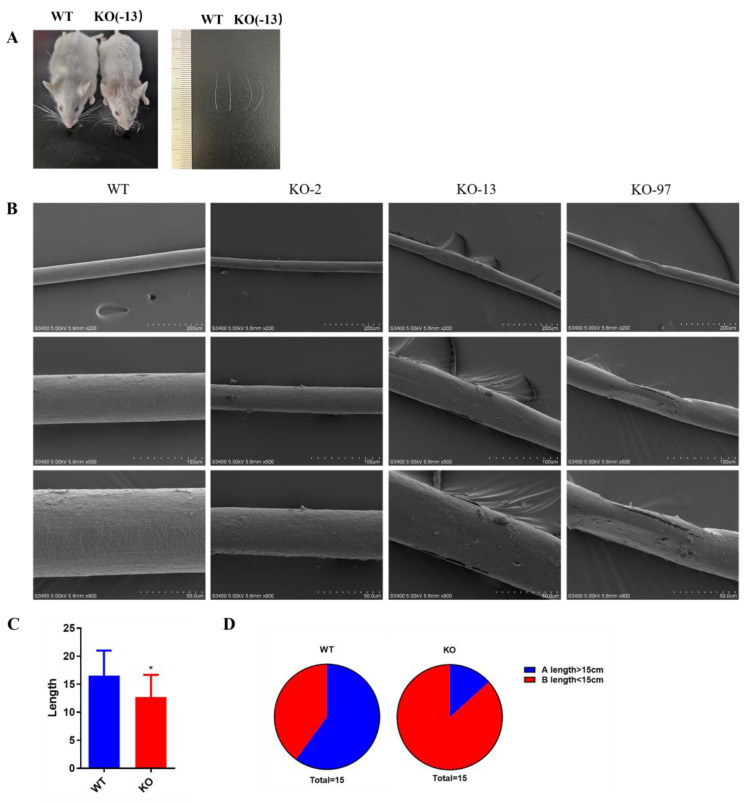
Structural changes in beards of Krt71–KO mice. (**A**) Schematic diagram of beard comparison of 12–week–old F1 Krt71–KO mice by photo. (**B**) SEM observation of beards from WT and Krt71–KO mice. SEM, scanning electron microscopy. Bars = 50 μm, 100 μm, 200 μm. (**C**) Statistical comparison of the length of beards in Krt71–KO mice and WT controls (*n* = 10, * *p* < 0.05). (**D**) Proportion of the length < 15 cm or >15 cm of the beards. The percent of the length < 15 cm is marked in red; the other percent of the length > 15 cm is marked in dark blue.

**Figure 6 cells-12-01781-f006:**
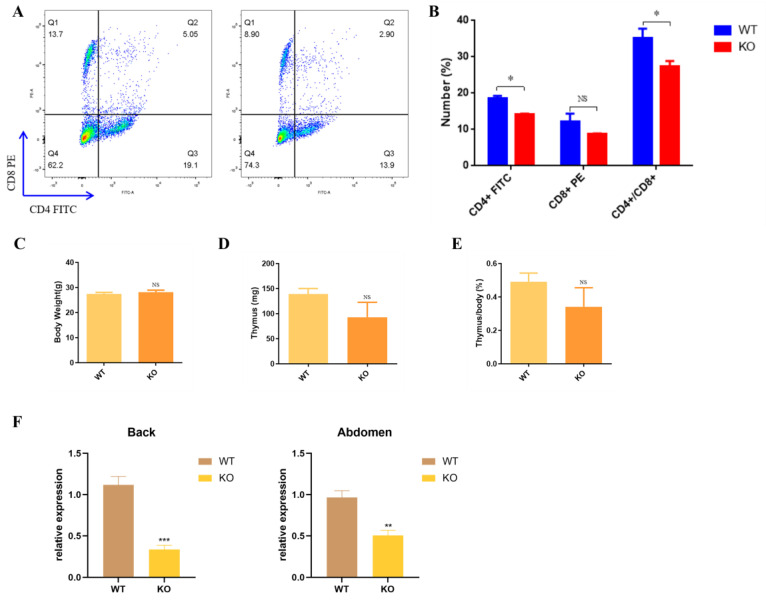
Immune level verification in Krt71–KO mice. (**A**) Representative flow cytometry results of peripheral blood lymphocytes in WT and Krt71–KO mice. (**B**) Summary of T cell populations in peripheral blood from three Krt71–KO mice in comparison with that from three WT mice. (**C**) Summary of body weight comparison between Krt71–KO and WT mice;(*n* = 3). (**D**) Summary of thymus weight comparison between Krt71–KO and WT mice; (*n* = 3). (**E**) Summary of body weight to thymus weight ratio for Krt71–KO and WT mice; (*n* = 3). (**F**) Gene expression of Foxn1 was determined by qRT–PCR in WT and Krt71–KO mice. (NS, no significance, * *p* < 0.05, ** *p* < 0.01, *** *p* < 0.001).

## Data Availability

The data presented in this study are available on request from the corresponding author.

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
