# Peer review of "Development of Woolly Hair and Hairlessness in a CRISPR−Engineered Mutant Mouse Model with KRT71 Mutations"

_cells, 2023, doi:10.3390/cells12131781_

Round 1

Reviewer 1 Report

(1) To clarify the novelty and significance of the current study, the authors should discuss over the commonality and difference in the phenotype (i.e. nude, immune-phenotype) between the established KO mice and previously reported KRT71 mutant mice, as well as the potential reasons for the difference.

(2) Discussion on the potential mechanism underlying the reduction of other KRT gene expression would be needed, which can be described in result section as the authors already did on MZF1, LPAR6, TGFa.

(3) Figure legend in Fig. 6F said that "KRT71" expression, but the main text said "Foxn1". Which is correct? If the latter is correct, why do the authors examine Foxn1 expression in skin? KO mice displayed the reduction of Foxn1, how can we interpret the results? The authors have to explain the rationale to assess Foxn1 expression as well as their interpretation on the data. If the former is correct (if the figure shows KRT71 expression), why the authors show that here? They already showed the KRT71 mRNA reduction (possibly by NMD) before (Fig. 1C). Again, all results need to be followed by their interpretation or the implication from them.

(4) I guess that Figure legend in Fig.2D said "FGF5", which has to be "KRT71". Or did the authors examine FGF5? if so why?

This manuscript has numerous grammar issues, which need to be addressed by getting editing help from someone with full professional proficiency in English.

Author Response

Repose to Reviewer 1 Comments:

(1) To clarify the novelty and significance of the current study, the authors should discuss over the commonality and difference in the phenotype (i.e. nude, immune-phenotype) between the established KO mice and previously reported KRT71 mutant mice, as well as the potential reasons for the difference.

Response: Thank you for your insightful and constructive comments and advice. The discussion on impact of miniaturization on the protein function were added in line 285-291 of the revised manuscript.

(2) Discussion on the potential mechanism underlying the reduction of other KRT gene expression would be needed, which can be described in result section as the authors already did on MZF1, LPAR6, TGFa.

Response: Thank you for your good suggestion. We have added the description in line 172-182 of the revised manuscript.

(3) Figure legend in Fig. 6F said that "KRT71" expression, but the main text said "Foxn1". Which is correct? If the latter is correct, why do the authors examine Foxn1 expression in skin? KO mice displayed the reduction of Foxn1, how can we interpret the results? The authors have to explain the rationale to assess Foxn1 expression as well as their interpretation on the data. If the former is correct (if the figure shows KRT71 expression), why the authors show that here? They already showed the KRT71 mRNA reduction (possibly by NMD) before (Fig. 1C). Again, all results need to be followed by their interpretation or the implication from them.

Response: Thank you for the helpful suggestions. The correct expression is "Foxn1" and the figure legend have changed in Fig.6F in line 272 of the revised manuscript. We explained the reason of the reduction of Foxn1 in the KO mice in line 258-262 of the revised manuscript.

(4) I guess that Figure legend in Fig.2D said "FGF5", which has to be "KRT71". Or did the authors examine FGF5? if so why?

Response: Thank you for the helpful suggestions. The correct expression is "Krt71" and the figure legend have changed in Fig.2D in line 208 of the revised manuscript.

Comments on the Quality of English Language

This manuscript has numerous grammar issues, which need to be addressed by getting editing help from someone with full professional proficiency in English.

Response: Thank you for the helpful suggestions. We have read the manuscript carefully, and the English writing has been modified by a native English speaker and a professional paper writing agency, which was marked in blue font in the revised manuscript.

Reviewer 2 Report

Dear authors, thank you very much for your interesting paper. I have only  minor editorial comments.

Please,

1. Throughout the text, make references to the literature according to the requirements of the journal.

2. Remove the italics in the legend for Figure 2.

3. Decide if it is appropriate to insert a discussion (line 167-170) in the results section.

Author Response

Repose to Reviewer 2 Comments:

Dear authors, thank you very much for your interesting paper. I have only minor editorial comments.

Please,

  1. Throughout the text, make references to the literature according to the requirements of the journal.

Response: Thank you for the helpful suggestions. We have changed the literature according to the requirements of the journal.

  1. Remove the italics in the legend for Figure 2.

Response: Thank you for your suggestion. We have removed the italics in the legend for Fig.2 in line 203-210 of the revised manuscript.

  1. Decide if it is appropriate to insert a discussion (line 167-170) in the results section.

Response: Thank you for your suggestion. We think it is appropriate for a discussion about the reason of the changes of other genes in line 172-182 in the results section.
